# Establishment of Reference Intervals of Hematological Parameters and Evaluation of Sex and Age Effect in the Miranda Donkey

**DOI:** 10.3390/ani13142331

**Published:** 2023-07-17

**Authors:** Grasiene Silva, Felisbina Queiroga, Madalena Ferreira, Daniela Andrade, Ana C. Silvestre-Ferreira

**Affiliations:** 1Departamento de Ciências Veterinárias, Universidade de Trás-os-Montes e Alto Douro (UTAD), 5000-801 Vila Real, Portugal; grasivet@hotmail.com (G.S.);; 2Centro de Ciência Animal e Veterinária (CECAV), Universidade de Trás-os-Montes e Alto Douro (UTAD), 5000-801 Vila Real, Portugal; 3Laboratório Associado para a Ciência Animal e Veterinária—AL4AnimalS, 5001-801 Vila Real, Portugal; 4AEPGA—Associação para o Estudo e Proteção Gado Asinino, M. Largo da Igreja, no. 48, 5225-011 Atenor, Portugal

**Keywords:** hematology, Miranda donkey, ProCyte Dx, reference intervals

## Abstract

**Simple Summary:**

The Miranda donkey is a Portuguese breed from northern Portugal that is considered endangered. Much research in different scientific areas has been conducted to help in these animals’ preservation. Knowledge of hematological reference intervals is important for characterizing the breed, identifying the health status of the animals and helping veterinarians in the diagnosis of diseases and the follow-up of patients. This study aimed to determine the hematological reference intervals for healthy Miranda donkeys and to evaluate the interference of sex and age in these parameters. Only age interfered in the hematological values and should be considered when interpreting the results. The results described here can be used to assess the health of animals and herds, guide the diagnosis of diseases and assist in the selection of healthy animals for reproduction, contributing to the preservation of the breed.

**Abstract:**

The Miranda donkey is an autochthonous Portuguese breed that is considered endangered. Several studies have been carried out on this breed, but to the authors’ best knowledge, no studies have been conducted on their clinical pathology. The aims of this study were to determine the hematological reference intervals (RIs) in healthy Miranda donkeys and to estimate the influence of age and sex. Blood samples from 75 clinically healthy animals were analyzed for 22 hematological parameters on the IDEXX ProCyte Dx, an automated hematology analyzer previously validated for the species. The RIs were estimated following the ASVCP guidelines with the Reference Value Advisor software. Regarding sex, no significant differences were found between groups. Regarding age, significant statistical differences (*p* < 0.05) were observed for red blood cells, red cell distribution width, white blood cells, lymphocytes, monocytes, platelets, plateletcrit (higher mean in young animals), mean corpuscular volume, mean corpuscular haemoglobin, neutrophils and eosinophils (higher mean in adults). The RIs described here can be used to evaluate and monitor the health status of animals and herds, as well as to guide diagnoses or select fit and healthy animals for reproduction, contributing to the preservation of the breed.

## 1. Introduction

The Miranda donkey is an autochthonous Portuguese breed from northern Portugal. It is currently classified by the Food and Agriculture Organization of the United Nations [1] as endangered due to its small number of animals. According to the latest survey, released in 2023, the current number of sexually mature Miranda donkeys consists of 726 animals, 652 females and 109 males [2]. The need to preserve this breed is linked to the conservation of its valuable genetic heritage together with cultural, historic and economical relevance, especially for family farming in the Planalto Mirandês region. Currently, after the implementation of government incentives and organizations for the defense of the breed, despite their continuous valuable contribution to agricultural activities, donkeys are also used in therapeutic activities, in leisure events, in ecotourism, as companions, in the production of milk for human consumption, in the manufacture of cosmetics, in the control of vegetation and in landscape maintenance [3]. The first steps in the preservation procedure of a given breed consist of studying its geographic distribution and number of animals together with its phenotypic and genotypic characteristics [4,5,6]. In addition, hematology studies that allow the identification of reference intervals (RIs) for hematological variables are important for characterizing a breed and have been used by scientists as a measure to potentiate the conservation of other autochthonous European breeds [7,8,9,10,11]. RIs allow for the evaluation of the health status of animals and herds and for monitoring the evolution of diseases and responses to therapy [12].

For a long time, veterinarians have used hematological RIs for horses as a guide for evaluating donkeys’ healthy status, which is not adequate, as there are considerable differences between both species [13,14]. Research on the hematological profiles of other donkey breeds shows that there are differences not only with horses but also differences between different donkey breeds, which demonstrates the relevance of studying this subject in each autochthonous breed [7,8,15]. Moreover, it is also relevant to study the impact of sex and age, as they can also influence blood parameters within the same population [16].

Research in several areas of veterinary medicine has been conducted on Miranda donkeys, but to the authors’ knowledge, there are no hematology studies. Determining a normal range for hematological parameters in an endangered population such as Miranda donkeys can be difficult due to the limited number of existing animals, but it is essential. The aims of this study were to determine the hematological RI for the Miranda donkey breed and to evaluate the possible influence of sex and age factors on these intervals.

## 2. Materials and Methods

### 2.1. Study Population

All the animals were born and lived in the same geographical area (41°42′ N 06°48′ W, 652 m of altitude) in the north of Portugal and were registered in the Associação para o Estudo e Proteção do Gado Asinino (AEPGA). Animals are frequently monitored by the association’s veterinarians and receive regular anthelmintic treatment, following a selective protocol [17,18]. Samples were collected as part of a prophylactic program developed for the breed, and no sample was collected on purpose for the study. All owners gave informed consent for the use of data and remaining blood samples from their animals. The study was approved by the ORBEA (Ethics Committee for Animal Welfare) by the Universidade Trás-os-Montes e Alto Douro (UTAD)—i467-e-CECAV-2022 and was carried out in spring.

To study the effect of age on the hematological profiles, the donkeys were divided into two groups: young (1–3 years old; *n* = 20) and adults (≥4 years old; *n* = 55). Regarding the effect of sex, the donkeys were grouped into males and females.

### 2.2. Sample Collection and Hematological Analyses

For the present study, the inclusion and exclusion criteria were established prior to sample collection, as follows: only healthy animals, routinely dewormed, were included. For each animal, the owners or caretakers stated their normal physical conditions and regular activity, any lack of signs of disease or any health problems in the previous 6 months. Pregnant and lactating mares were also included [11,19]. Animals that, after evaluation by a veterinarian, were deemed unhealthy, or that received any medication in the previous 6 months, as well as samples from animals that were excited or agitated at the time of sampling, were excluded.

All the animals were considered clinically healthy after anamnesis and physical examination by veterinarians, and they were handled carefully to reduce any possible effects of stress on the analyzed parameters.

Samples were collected as part of an annual prophylactic program developed for the breed. Blood samples were obtained in the morning period from non-fasted animals, via jugular vein puncture using a 21 gauge needle (BD Vacutainer PrecisionGilde) into K3EDTA vacuum tubes (BD Vacutainer) with a capacity of 4 mL of blood. Blood samples were homogenized and processed until 1 h after collection with the IDEXX ProCyte Dx automatic hematology analyzer for veterinary use, which operates using three key technologies: laser flow cytometry, optical fluorescence and laminar flow impedance. For hemoglobin measurement, the SLS-hemoglobin method was applied. This equipment was previously validated for the species [20] and was used following the manufacturer’s instructions. Samples with agglutination signs were rejected. For this, a small laboratory structure was set up at AEPGA.

The following parameters were obtained: red blood cell count (RBC), hematocrit (HCT), hemoglobin concentration (HB), mean corpuscular volume (MCV), mean corpuscular hemoglobin (MCH), mean corpuscular hemoglobin concentration (MCHC), red cell distribution width (RDW), white blood cell count (WBC), WBC differential total count and percentage of neutrophils (NEU), lymphocytes (LYM), monocytes (MONO), eosinophils (EOS), basophils (BASO), platelet count (PLT), mean platelet volume (MPV), platelet distribution width (PDW), and plateletcrit (PCT).

### 2.3. Statistical Analyses

The descriptive statistics, RIs and confidence interval (CI) limits at 90% were established with the Reference Value Advisor v2.1 [21]. Data normality was assessed using the Anderson–Darling test, and the presence of outliers was determined using the Tukey and Dixon–Reed method. Depending on the distribution, the parametric or robust method with or without Box–Cox transformation was applied, and the RIs and 90% confidence intervals (CI) for the lower and upper limits were calculated [22].

The calculations used to investigate the eventual significant differences in relation to the effect of the sex and age groups on hematological parameters were performed with SPSS program version 27. The Shapiro–Wilk test was performed to test sample normality, and then samples were subjected to an analysis of variance (ANOVA) to investigate the influence of sex and age. Differences were considered significant when *p* < 0.05.

## 3. Results

### 3.1. Descriptive Analysis and Reference Intervals for the Total Number of Animals

A total of 75 healthy Miranda donkeys (42 females with a mean age of 8 ± 6.09 years ranging from 1 to 25 years, and 33 males with a mean age of 8.23 ± 5.14 years ranging from 1 to 18 years) were used in the study. A total of twenty-two hematological parameters were measured. All data are expressed as the mean, median, standard deviation (SD), lowest and highest values (Min–Max), RI, 90% CI for the lower limit and 90% CI for the upper limit. The respective results for the whole population are described in Table 1.

### 3.2. Influence of Sex and Age

Based on the Shapiro–Wilk test, it was determined that the parameters followed a normal distribution (*p* > 0.05).

Considering sex (33 males vs. 42 females), no significant differences were found between groups for any of the analyzed parameters (*p* > 0.05).

Considering age (20 young vs. 55 adult animals), differences were found for RBC, RDW, WBC, LYM (%), LYM, MONO, PLT, PCT (higher average in young), VCM, MCH, NEU (%), EOS (%) and EOS (higher mean in adults) (Table 2).

Taking into consideration that age had an impact on several hematological parameters, RIs were calculated for young and adult animals and are presented in Table 3 and Table 4, respectively.

## 4. Discussion

For a long time, veterinarians have used equine reference values in donkey medicine due to similarities between both species; however, this practice is not recommended, because there are several differences between the species [13,14].

Currently, studies that determine the hematological RIs in donkey breeds have increased, and some breeds have known reference values, such as the Balkan [9], Catalan donkey [7], Herzegovina [23], Martina Franca [11], Pêga [24] and Ragusana [19].

This study is the first to determine hematological RIs for the Portuguese Miranda donkey breed and to evaluate the influence of sex and age on these parameters. Comparative studies of hematological RIs are difficult to perform because RIs are influenced by different methodologies, equipment and statistical methods [16]. In our study, the analyses were performed with the IDEXX ProCyte Dx, an automatic hematology analyzer previously validated for the species by our research group. IDEXX ProCyte Dx has good cell count efficiency in this species and may be used for future research on other donkey breeds [20].

No significant differences were found between males and females. This result agrees with other studies that have also not found significant differences between sex in Catalan donkeys [7] and in mixed-breed donkeys [25,26]. However, other authors have reported significant differences between sexes for several hematological parameters in donkeys from France [8], Serbia [9] and Bósnia [10]. In our study, it is possible that sex did not influence the hematological parameters because most males were castrated. Generally, males of this breed are castrated for reproductive reasons, as a way of selecting and reproducing only animals with characteristic breed standards. Another reason is that many breeders are elderly and, to facilitate handling, prefer castrated males, as the animals are more docile and less temperamental [5]. In different species, the hematological profile seems to differ between castrated and intact animals [27,28,29]. This is possibly due to the effects of testosterone on hematopoiesis [30]. A study conducted with goats found a significant difference in hematological parameters for MPV, HB, RBC, WBC and MCH, which decreased after castration, and for MCH and platelets, which increased [29]. The influence of testosterone has also been studied in other species, such as rabbits and rats [27,28]. In donkeys, there was a study conducted in a herd in which all animals were castrated, but there have been none comparing castrated and non-castrated males. It is necessary that future research be carried out on this species with a larger number of non-castrated males to evaluate the influence of sex and/or of testosterone [31]. 

Regarding age, significant differences were found in some parameters, which is in accordance with research carried out with donkeys of other European breeds, such as the Catalan from Spain [7], Cotentin and Normande from France [8], Ragusana and Martina Franca from Italy [11,19] and Balkan from Serbia [9]. The younger group had higher averages for RBC, and this result has been previously described [8,19,32,33]. In general, it is accepted that younger animals have a higher RBC than that of adults, because, with advancing age, there is a decrease in the medullary response [34]. Another explanation could be the more stressed temperament of younger donkeys, which are more stressed during collection and thus may suffer splenocontraction with the release of RBC into circulation [8]; however, samples from animals that were excited or agitated at the time of sampling were excluded from this study.

Chronic diseases also can cause a decrease in RBC [12], but the adult animals in our research were healthy and did not have clinical signs or a history of chronic diseases. Therefore, we believe that the decline in the values of RBC with age was caused by decreased medullary activity. The RDW was also higher in young animals, which is in agreement with the results found in Ragusana [19] and Martina Franca [11] donkeys.

MCV increased significantly with age in our study, as previously described in other studies with healthy donkeys [7,8,24,35] and horses [36]. In horses, higher MCV values may be associated with a regenerative response to anemia, but in the absence of anemia, the larger size of red blood cells may reflect changes derived from the dynamics of red blood cell maturation [36]. Research carried out on humans related the increase in MCV to an alteration in the process of cell division caused by a deficiency of vitamin B_12_ or erythrocyte folate [37]. Although horses do not need a dietary supplement of vitamin B_12_, because they can produce it through microbial fermentation in the large intestine, some situations can cause a reduction, such as intensive training, exercise or confinement [36]. In donkeys, the causes for this increase were not discussed, but in a study with horses, it was suggested that this decrease in vitamin B_12_ could be due to the lower digestive capacity of older animals [38]. The progressive increase in MCH in Miranda donkeys likely was the result of the larger size of the erythrocytes and has also been observed in old donkeys of various breeds [8,32,35].

The decrease in leukocytes with advancing age observed in our study was one of the parameters that we found to be most influenced by age, as described previously [7,8,11,33]. The WBC in healthy donkeys depends on several factors, such as age, nutritional status, pregnancy and lactation [7,39]. In healthy older animals, it can be related to declines in immunocompetence [31]. 

For the differential leukocyte count, we found higher values for NEU, LYM and MONO, which is similar to what has been described in previous studies [7,9,11,18]. In horses, the decrease in lymphocytes due to immunosenescence was discussed by McFarlane [40]. In contrast, EOS increased in adult animals. According to other authors, the number of EOS tends to increment with age, likely due to the progressive exposure of animals during their life [41].

Regarding platelets, in this study, adult donkeys were found to have the lowest PLT and PCT. This corresponds to findings by Foch [7], Pitel [8] and Stanišić [9], who reported that age determines progressive decreases in platelet count, possibly due to a decrease in production with aging. PLT can vary according to age, breed, exercise, training and reproductive status. In the case of disease, there may be thrombocytosis in response to inflammatory cytokines or thrombocytopenia due to mechanisms such as reduced thrombopoiesis, increased peripheral destruction of platelets, spleen sequestration and the loss of platelets due to idiopathic origin [42].

## 5. Conclusions

The Miranda donkey constitutes a Portuguese genetic heritage that must be protected, and efforts are still needed to increase the number of individuals, especially those of reproductive and fertile age, in order to avoid the extinction of the breed. In recent years, the interest of researchers has increased, and the number of publications on the breed has grown. However, some scientific areas, such as clinical pathology, have been neglected. 

The RIs described here can be used to assess and monitor the health status of animals and herds, as well as to guide diagnoses or select fit and healthy animals for reproduction, contributing to the preservation of the breed. It is important to emphasize that age can influence the hematological results and therefore must be considered by professionals when interpreting tests.

## Figures and Tables

**Table 1 animals-13-02331-t001:** Hematological reference intervals for the population of Miranda donkeys included in the study (*n* = 75).

Parameters/Units	*n*	Mean ± SD	Median	Min–Max	RI	LRL 90% CI	URL 90% CI
RBC (M/μL)	75	5.2 ± 0.7	5.1	3.86–6.87	4.0–6.8	3.9–4.3	6.4–6.9
HCT (%)	75	30.2 ± 3.5	29.8	24.1–40.6	24.5–38.4	24.1–25.3	36.6–40.6
HB (g/dL)	75	10.4 ± 1.1	10.3	8.2–13.3	8.5–13.3	8.2–8.8	12.4–13.3
MCV (fL)	75	58.1 ± 3.9	58.3	49.5–66.9	50.2–66.5	49.5–52.0	64.1–66.9
MCH (pg)	75	20.0 ± 1.2	20.0	17.4–23.5	17.5–23.0	17.4–18.1	21.9–23.5
MCHC(g/dL)	75	34.4 ± 0.9	34.5	32.2–36.4	32.5–36.4	32.2–32.8	35.6–36.4
RDW (%)	75	21.8 ± 1.6	21.5	19.4–25.6	19.4–25.4	19.4–19.6	24.6–25.6
WBC (K/μL)	75	8.2 ± 2.0	8.1	4.89–13.45	5.0–12.2	4.9–5.3	11.4–13.5
NEU (%)	75	44.1 ± 6.4	43.7	31.8–66.5	31.9–59.4	31.8–34.5	53.4–66.5
LYM (%)	75	41.9 ± 7.2	41.3	25.4–55.2	28.2–55.0	25.4–32.5	53.7–55.2
MONO (%)	74	5.5 ± 0.8	5.4	4.1–7.6	4.2–7.4	4.1–4.4	6.9–7.6
EOS (%)	75	8.0 ± 3.3	7.4	2.2–16.5	2.7–15.6	2.2–3.8	14.2–16.5
BASO (%)	75	0.4 ± 0.4	0.4	0.0–1.6	0.0–1.6	0.0–0.1	1.2–1.6
NEU (K/μL)	75	3.6 ± 1.1	3.5	1.81–7.81	2.0–6.9	1.8–2.3	5.4–7.8
LYM (K/μL)	75	3.4 ± 1.1	3.2	1.69–5.97	1.7–5.7	1.7–1.9	5.3–6.0
MONO (K/μL)	75	0.5 ± 0.1	0.4	0.23–0.84	0.2–0.8	0.2–0.3	0.7–0.8
EOS (K/μL)	75	0.6 ± 0.3	0.6	0.19–1.4	0.3–1.3	0.2–0.3	1.1–1.4
BASO (K/μL)	75	0.0 ± 0.0	0.0	0.0–0.12	0.0–0.1	0.0–0.0	0.1–0.1
PLT (K/μL)	75	221.1 ± 56.0	228.0	74.0–344.0	93.8–341.3	74.0–139.0	298.3–344.0
MPV (fL)	75	6.1 ± 0.5	6.0	5.3–7.9	5.3–7.6	5.3–5.4	7.1–7.9
PDW (%)	75	6.8 ± 0.7	6.7	5.8–8.7	5.8–8.7	5.8–6.0	8.1–8.7
PCT (%)	75	0.1 ± 0.0	0.1	0.03–0.19	0.0–0.2	0.0–0.1	0.2–0.2

RBC: red blood cell, HCT: hematocrit, HB: hemoglobin concentration, MCV: mean corpuscular volume, MCH: mean corpuscular hemoglobin, MCHC: mean corpuscular hemoglobin concentration, RDW: red cell distribution width, WBC: white blood cell, NEU: neutrophils, LYM: lymphocytes, MONO: monocytes, EOS: eosinophils, BASO: basophils, PLT: platelets, MPV: mean platelet volume, PDW: platelet distribution width, PCT: plateletcrit, SD: standard deviation, Min: minimum, Max: maximum, RI: reference interval, LRL: lower reference limit, URL: upper reference limit, CI: confidence interval.

**Table 2 animals-13-02331-t002:** Effect of age on hematological parameters in Miranda donkeys.

Effect of Age			
Parameters/Units	Young *(n* = 20)	Adults (*n* = 55)	*p*-Value
	Mean ± SD	Mean ± SD	Mean
RBC (M/μL)	5.50 ± 0.72	5.10 ± 0.61	0.022
MCV (fL)	54.62 ± 3.71	59.39 ± 3.13	<0.001
MCH (pg)	18.91 ± 1.14	20.4 ± 0.93	<0.001
RDW (%)	22.75 ± 1.54	21.39 ± 1.49	0.001
WBC (K/μL)	9.04 ± 1.71	7.82 ± 1.96	0.017
NEU (%)	39.04 ± 4.30	45.95 ± 6.08	<0.001
LYM (%)	49.00 ± 4.08	39.38 ± 6.27	<0.001
EOS (%)	5.69 ± 2.08	8.82 ± 3.23	<0.001
LYM (K/μL)	4.40 ± 0.76	3.08 ± 0.94	<0.001
MONO (K/μL)	0.53 ± 0.15	0.42 ± 0.12	0.003
EOS (K/μL	0.50 ± 0.16	0.68 ± 0.27	0.008
PLT (K/μL)	261.65 ± 46.80	206.41 ± 51.89	<0.001
PCT (%)	0.13 ± 0.02	0.10 ± 0.02	<0.001

RBC: red blood cell, MCV: mean corpuscular volume, MCH: mean corpuscular hemoglobin, RDW: red cell distribution width, WBC: white blood cell, NEU: neutrophils, LYM: lymphocytes, MONO: monocytes, EOS: eosinophils, PLT: platelets, PCT: plateletcrit, SD: standard deviation.

**Table 3 animals-13-02331-t003:** Hematological reference intervals for young healthy Miranda donkeys (*n* = 20).

Parameters/Units	*n*	Mean ± SD	Median	Min–Max	RI	LRL 90% CI	URL 90% CI
RBC (M/μL)	20	5.5 ± 0.7	5.3	4.11–6.87	4.1–7.3	3.8–4.5	6.5–7.9
HCT (%)	20	30.0 ± 3.7	28.9	24.6–37.7	24.0–41.1	23.0–25.3	35.8–48.9
HB (g/dL)	20	10.4 ± 1.3	10.0	8.5–13.3	8.2–14.0	7.8–8.6	12.2–15.6
MCV (fL)	19	54.0 ± 2.4	54.2	49.5–59.1	48.8–59.2	47.3–50.4	57.5–60.8
MCH (pg)	19	18.7 ± 0.7	19.0	17.4–19.6	17.3–20.2	*	*
MCHC(g/dL)	20	34.6 ± 0.8	34.9	32.7–36.0	33.1–36.7	32.3–33.8	35.9–37.2
RDW (%)	20	22.8 ± 1.5	22.5	19.5–25.4	19.1–25.9	18.0–20.3	24.8–26.7
WBC (K/μL)	20	9.0 ± 1.7	8.7	6.42–12.06	5.4–12.7	4.1–6.5	11.6–14.0
NEU (%)	20	39.0 ± 4.3	39.2	31.8–48.0	30.2–48.7	28.1–32.9	45.6–51.5
LYM (%)	20	49.0 ± 4.1	49.7	40.7–55.2	38.2–56.4	33.0–42.9	54.4–57.9
MONO (%)	20	5.8 ± 0.9	5.5	4.7–7.6	4.4–8.8	4.1–4.6	7.1–11.0
EOS (%)	20	5.7 ± 2.1	5.4	2.2–9.9	1.9–10.7	1.4–2.8	9.0–12.4
BASO (%)	20	0.5 ± 0.5	0.4	0.0–1.6	*	*	*
NEU (K/μL)	20	3.6 ± 1.0	3.4	2.36–5.78	1.5–5.6	0.9–2.2	5.0–6.2
LYM (K/μL)	20	4.4 ± 0.8	4.1	3.1–5.97	3.0–6.5	2.8–3.3	5.4–7.4
MONO (K/μL)	20	0.5 ± 0.2	0.5	0.33–0.83	1.6–5.1	1.4–1.9	4.0–5.9
EOS (K/μL)	20	0.5 ± 0.2	0.5	0.19–0.8	0.1–0.8	0.0–0.3	0.7–0.9
BASO (K/μL)	20	0.0 ± 0.0	0.0	0.0–0.1	0.0–0.2	0.0–0.0	0.1–0.3
PLT (K/μL)	20	261.7 ± 46.8	272.0	143.0–344.0	144.2–350.4	94.0–194.6	324.3–376.0
MPV (fL)	20	6.1 ± 0.5	6.1	5.3–7.1	5.1–7.2	4.9–5.4	6.8–7.6
PDW (%)	20	6.9 ± 0.7	6.7	5.8–8.6	5.8–8.7	5.5–6.0	7.8–9.7
PCT (%)	20	0.1 ± 0.0	0.1	0.11–0.17	0.1–0.2	0.1–0.1	0.2–0.2

RBC: red blood cell, HCT: hematocrit, HB: hemoglobin concentration, MCV: mean corpuscular volume, MCH: mean corpuscular hemoglobin, MCHC: mean corpuscular hemoglobin concentration, RDW: red cell distribution width, WBC: white blood cell, NEU: neutrophils, LYM: lymphocytes, MONO: monocytes, EOS: eosinophils, BASO: basophils, PLT: platelets, MPV: mean platelet volume, PDW: platelets distribution width, PCT: plateletcrit, SD: standard deviation, Min: minimum, Max: maximum, RI: reference intervals, LRL: lower reference limit, URL: upper reference limit, CI: confidence interval, *: non computable

**Table 4 animals-13-02331-t004:** Hematological reference intervals for adult healthy Miranda donkeys (*n* = 55).

Parameters/Units	*n*	Mean ± SD	Median	Min–Max	RI	LRL 90% CI	URL 90% CI
RBC (M/μL)	55	5.1 ± 0.6	5.0	3.86–6.76	3.9–6.6	3.9–4.2	6.2–6.8
HCT (%)	55	30.3 ± 3.4	29.9	24.1–40.6	24.3–39.6	24.1–25.4	36.1–40.6
HB (g/dL)	55	10.4 ± 1.1	10.3	8.2–13.3	8.4–13.2	8.2–8.9	12.6–13.3
MCV (fL)	55	59.4 ± 3.1	59.6	52.3–66.4	52.7–66.2	52.3–54.1	63.9–66.4
MCH (pg)	55	20.4 ± 0.9	20.4	18.5–23.5	18.6–22.9	18.5–18.9	21.7–23.5
MCHC(g/dL)	55	34.4 ± 0.9	34.4	32.2–36.4	32.3–36.4	32.2–32.8	35.7–36.4
RDW (%)	55	21.4 ± 1.5	20.9	19.4–25.6	19.4–25.3	19.4–19.6	24.4–25.6
WBC (K/μL)	55	7.8 ± 2.0	7.8	4.89–13.45	4.9–12.8	4.9–5.3	11.0–13.5
NEU (%)	55	46.0 ± 6.1	45.7	33.4–66.5	33.8–63.3	33.4–37.5	53.6–66.5
LYM (%)	55	39.4 ± 6.3	38.2	25.4–55.0	26.6–54.5	25.4–31.2	49.8–55.0
MONO (%)	55	5.4 ± 0.9	5.4	4.1–8.6	4.1–8.1	4.1–4.3	6.8–8.6
EOS (%)	55	8.8 ± 3.2	8.1	2.8–16.5	3.3–16.1	2.8–4.6	14.4–16.5
BASO (%)	55	0.4 ± 0.3	0.4	0.0–1.6	0.0–1.4	0.0–0.1	0.9–1.6
NEU (K/μL)	55	3.6 ± 1.1	3.5	1.81–7.81	1.9–7.4	1.8–2.2	5.6–7.8
LYM (K/μL)	55	3.1 ± 0.9	3.0	1.69–5.55	1.7–5.5	1.7–1.8	4.8–5.6
MONO (K/μL)	55	0.4 ± 0.1	0.4	0.23–0.84	0.2–0.8	0.2–0.3	0.6–0.8
EOS (K/μL)	55	0.7 ± 0.3	0.7	0.26–1.4	0.3–1.4	0.3–0.3	1.2–1.4
BASO (K/μL)	55	0.0 ± 0.0	0.0	0.0–0.12	0.0–0.1	0.0–0.0	0.1–0.1
PLT (K/μL)	55	206.4 ± 51.9	212.0	74.0–310.0	82.8–308.4	74.0–139.0	283.0–310.0
MPV (fL)	55	6.1 ± 0.6	5.9	5.3–7.9	5.3–7.8	5.3–5.4	7.1–7.9
PDW (%)	55	6.8 ± 0.7	6.6	5.8–8.7	5.9–8.7	5.8–6.1	8.0–8.7
PCT (%)	54	0.1 ± 0.0	0.1	0.03–0.15	0.0–0.1	0.0–0.1	0.1–0.2

RBC: red blood cell, HCT: hematocrit, HB: hemoglobin concentration, MCV: mean corpuscular volume, MCH: mean corpuscular hemoglobin, MCHC: mean corpuscular hemoglobin concentration, RDW: red cell distribution width, WBC: white blood cell, NEU: neutrophils, LYM: lymphocytes, MONO: monocytes, EOS: eosinophils, BASO: basophils, PLT: platelets, MPV: mean platelet volume, PDW: platelet distribution width, PCT: plateletcrit, SD: standard deviation, Min: minimum, Max: maximum, RI: reference intervals, LRL: lower reference limit, URL: upper reference limit, CI: confidence interval.

## Data Availability

All the data supporting the results are included in the manuscript. The dataset is available from the corresponding author on reasonable request.

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
