# Peer review of "Establishment of Reference Intervals of Hematological Parameters and Evaluation of Sex and Age Effect in the Miranda Donkey"

_animals, 2023, doi:10.3390/ani13142331_

Round 1
Reviewer 1 Report
Major revision
The submission is related to reference intervals of Miranda’s donkey. It is a very interesting and important manuscript, however, as the authors are proposing reference intervals for this specie and in order to accept this manuscript for publication in Animals, I recommend the following corrections:
1. Materials and Methods: Study population – considering a reference interval (RI) study, the animals should be higid. According to authors report, there are no RI for this specie, however there are for other primates and could be used for comparison. This statement is important because we must be certain that all included animals are healthy. Also, I strongly recommend that at least serum biochemistry should be done at these samples and compare with other primate species. Another question: Were these animals evaluated by a vet during the blood sampling? It should be clear at mat and meths.
2. Discussion: I strongly recommend discuss the results in relation to primates (human and non human) instead of horses. Horse hematology is different than primates, and also dogs and cats for example. Discussion must be improved.
Based on these statements, major revisions are necessary.
Author Response
Referee 1 comments:
The submission is related to reference intervals of Miranda’s donkey. It is a very interesting and important manuscript, however, as the authors are proposing reference intervals for this specie and in order to accept this manuscript for publication in Animals, I recommend the following corrections:
1. Materials and Methods: Study population – considering a reference interval (RI) study, the animals should be higid. According to authors report, there are no RI for this specie, however there are for other primates and could be used for comparison. This statement is important because we must be certain that all included animals are healthy. Also, I strongly recommend that at least serum biochemistry should be done at these samples and compare with other primate species. Another question: Were these animals evaluated by a vet during the blood sampling? It should be clear at mat and meths.
2. Discussion: I strongly recommend discuss the results in relation to primates (human and non human) instead of horses. Horse hematology is different than primates, and also dogs and cats for example. Discussion must be improved.
Authors answer: The authors thanks the referee comments, however we believe that there must be some kind of mistake because Miranda's donkey (Equus asinus) is not a primate but an equine, so it makes, for us, perfect sense to compare our results with those of horses. We hope the referee understands our decision of not compare results with primates.
SPECIFIC COMMENTS
Referee comment: Were these animals evaluated by a vet during the blood sampling? It should be clear at mat and meths.
Authors answer: the authors added this information in results, lines 94-95.
Reviewer 2 Report
GENERAL COMMENTS
-The manuscript reports the results of haematological analyses performed on blood samples from an autochtonous donkey breed (Miranda donkey), putting in evidence the possible effects of sex and age.
-The paper is -in my opinion- very interesting, since it brings out several characteristics of a local breed, of paramount importance for the maintaining/recovery of local and/or endangered breeds.
-The paper is complete, well-written, and clear. I strongly appreciate the introduction of LRL and URL 90% CI for reference limits, as generally recommended.
For these reasons, I recommend the acceptance of the present manuscript for publication on Animals (minor revisions).
SPECIFIC COMMENTS
-Statistical analysis - Maybe the specification of Normal/non-Normal distributions could complete the information about parameters.
-Table 2 - The P-value on the heading seems to be an oversight; maybe the simple indication of "P-value" could clarify the information.
-Did the authors take into account the possibility to perform a two-way ANOVA in order to consider the coexistence of age/sex contemporarily?
-Discussion - The authors report the sex differences between parameters, specifying that some males have been castrated; maybe the integration of this aspect could suggest further research ways on this interesting field. I suggest a brief report on the differences between castrated/non-castrated males (even not supported by inferential statistics).
Author Response
Referee 2 comments:
-The manuscript reports the results of haematological analyses performed on blood samples from an autochtonous donkey breed (Miranda donkey), putting in evidence the possible effects of sex and age.
-The paper is -in my opinion- very interesting, since it brings out several characteristics of a local breed, of paramount importance for the maintaining/recovery of local and/or endangered breeds.
-The paper is complete, well-written, and clear. I strongly appreciate the introduction of LRL and URL 90% CI for reference limits, as generally recommended.
For these reasons, I recommend the acceptance of the present manuscript for publication on Animals (minor revisions).
Authors answer: The authors highly appreciated and thanks the referee comments
SPECIFIC COMMENTS
Referee comment: -Statistical analysis - Maybe the specification of Normal/non-Normal distributions could complete the information about parameters.
Authors answer: the authors added this information in results, lines 148-149.
Referee comment: -Table 2 - The P-value on the heading seems to be an oversight; maybe the simple indication of "P-value" could clarify the information.
Authors answer: authors thanks referee comment. It was a typing error already corrected.
Referee comment: -Did the authors take into account the possibility to perform a two-way ANOVA in order to consider the coexistence of age/sex contemporarily?
Authors answer: In fact, we did not. Our group has been publishing several manuscripts in this topic and never investigated the join effect of 2 factors simultaneously. Although the authors find the suggestion very interesting, given the number of analysed factors (n=22 factors analysed), a considerable additional statistical analysis would be needed and a new table for sure would be required for inclusion of new values, which would make this manuscript considerably more complex to read. Also, because the already published studies in other autochthonous breeds only present the values with a similar statistical analysis, we think it is easier for authors make comparative studies. We hope the referee understands our decision.
Referee comment:-Discussion - The authors report the sex differences between parameters, specifying that some males have been castrated; maybe the integration of this aspect could suggest further research ways on this interesting field. I suggest a brief report on the differences between castrated/non-castrated males (even not supported by inferential statistics).
Authors answer: the authors added this information in discussion, lines 201-210.
Round 2
Reviewer 1 Report
I accept the corrections made by the authors.